Variation in genetics, morphology, and recruitment of the invasive barnacle Amphibalanus eburneus (Gould, 1841) in the southern Korean peninsula

Kim Jeongho 1
Ubagan Michael 1 2
Kwon Soyeon 2
Kim Il-Hoi 3
Shin Sook 1 2 shins@syu.ac.kr
1 Marine Biological Resources Institute, Sahmyook University , Seoul , South Korea
2 Department of Animal Biotechnology and Resource, College of Science and Technology, Sahmyook University , Seoul , South Korea
3 Department of Biology, College of Natural Science, Gangneung-Wonju National University , Gangneung , South Korea
Reimer James
Electronic publication date: 2022 Sep 2
Publication date: 2022
Volume: 10
Electronic Location ID: e14002
Received 2021 Nov 3; Accepted 2022 Aug 13
Copyright: © 2022 Kim et al.
Copyright year: 2022
Copyright holder: Kim et al.
License: This is an open access article distributed under the terms of the Creative Commons Attribution License, which permits unrestricted use, distribution, reproduction and adaptation in any medium and for any purpose provided that it is properly attributed. For attribution, the original author(s), title, publication source (PeerJ) and either DOI or URL of the article must be cited.
License URL: https://creativecommons.org/licenses/by/4.0/

Keywords: Cirripedia, Biological invasion, Polymorphism, Phenotype, Operculum, Harmful, Local adaptaton

Funding: Improvement of Management Strategies on Marine Ecosystem Disturbing and Harmful Organisms 20190518 Monitoring Survey on the Distribution of Disturbing and Harmful Benthos in the Marine Ecosystem (2022) Ministry of Oceans and Fisheries This research was supported by the “Improvement of management strategies on marine ecosystem disturbing and harmful organisms (No. 20190518)” and “Monitoring survey on the distribution of disturbing and harmful benthos in the marine ecosystem (2022)” funded by the Ministry of Oceans and Fisheries. The funders had no role in study design, data collection and analysis, decision to publish, or preparation of the manuscript.

==============================
The ivory barnacle Amphibalanus eburneus is a marine crustacean, which presents near-cosmopolitan distribution due to extensive introduction and exhibits a wide spectrum of phenotypic variation. To elucidate geographical differentiation among populations through invasion, we investigated variation in genetic structure, shell morphology, and recruitment pattern for A. eburneus, from the southern Korean Peninsula where it has been established since the late 1980s. We selected samples from four populations in corresponding ecologically-relevant regions representing all surrounding South Korean waters. From these we amplified the mitochondrial genetic marker cytochrome oxidase subunit I (COI) from 57 individuals and performed a populational genetic analyses with 11 additional GenBank sequences to evaluate population structure. To examine morphological variation, we applied two-dimensional landmark-based geometric morphometrics to the scutum and tergum for 148 and 151 individuals, respectively. Furthermore, we estimated the density of year-old individuals in the field to compare recruitment responses among localities. We detected 33 haplotypes among the four locations belonging to three distinct clades based on moderate intraspecific pairwise genetic distance (≥3.5%). The haplotypes in these clades were not locality-specific in their distribution. In contrast, we did detect interpopulation variation in opercular shape and morphospace structure, and one population could be separated from the rest based on its distinct tergum morphotype alone. This morphologically distinct population was also differentiated by displaying the lowest mean recruitment density. Our results indicate that although there is no relationship between molecular variation in the COI gene and geographic regions in South Korea, association with locality for operculum morphology, and recruitment response suggest ecological adaptation by this barnacle in a new habitat.

Introduction

Barnacles are often dominant inhabitants of coastal ecosystems with some species that are transportable and that have spread across the globe (Gilg et al., 2010). The cosmopolitan distribution of a number of species may be attributed to human-mediated translocation recently and over the past centuries, primarily via ballast water and ship-hulls, serving as the vectors for planktonic larvae and sessile adults, respectively (Zardus & Hadfield, 2005; Davidson et al., 2009; Carlton, Newman & Pitombo, 2011). Widespread barnacle species with long-range dispersal and massive population sizes often exhibit phenotypic variations across geographic populations. Specifically, morphological variations in fitness-related physical traits in response to heterogeneous environments has been commonly documented, which allow for the comprehensive study of adaptive evolution. For instance, wave exposure levels drive morphological variations in the calcareous structures and chitinous exoskeleton of individuals to protect the soft body inside (Pentcheff, 1991). Similarly, cirrus morphology varies with wave exposure to enhance particle capture efficiency (Arsenault, Marchinko & Palmer, 2001), alternating between shorter and thicker in high exposure situations and longer and thinner forms in wave-protected zones (Marchinko & Palmer, 2003; Li & Denny, 2004; Miller, 2007; López et al., 2010). Furthermore, barnacles with bathymetric distribution often exhibit morphological variations in the cirri, being longer in individuals inhabiting the upper intertidal zone than in those inhabiting the middle-lower intertidal areas (Chan & Hung, 2005). Finally, the density at which individuals recruit and grow is another significant factor, resulting in competition for space and mating, which further induces shell elongation (Barnes & Powell, 1950; Bertness, Gaines & Yeh, 1998) and variations in penis length (Hoch, 2008).

The ivory barnacle Amphibalanus eburneus (Gould, 1841) belonging to the family Balanidae is native to the east coast of the USA, distributed from Nova Scotia to Florida, including the Caribbean and Gulf of Mexico (Kaplan, 1988). However, the species has now become nearly cosmopolitan due to extensive ship fouling (Henry & McLaughlin, 1975; Larsen, 1985). Until recently, the range of A. eburneus distribution was limited to European seas (Molnar et al., 2008; Jaberimanesh et al., 2019; Osca & Crocetta, 2020), Pacific Ocean (Henry & McLaughlin, 1975; Iwasaki, 2018), the Indian Ocean (Biccard & Griffiths, 2016), and the Canadian Arctic (Chan, MacIsaac & Bailey, 2015). Now, however, A. eburneus has become a harmful invader in many countries, inflicting extensive ecological and industrial damage due to a suite of distinct traits (Visscher, 1927; Haderlie, 1984). First, A. eburneus is euryhaline, exhibiting a preference for waters with salinity ranging between 15 and 20 ppt, which allows the species to endure drastic differences in salinity across diverse geographic habitats (Bacon, 1971; Jaberimanesh et al., 2019). Second, this species undergoes larval development across a range of temperatures (Scheltema & Williams, 1982); as such, the duration from the nauplius to the cyprid stage may vary from 7 to 13 days (Costlow & Bookhout, 1957). Given these high competitive abilities, A. eburneus is an excellent model to study genetic, morphological, and ecological variation across geographic populations accompanying biological invasion.

The Korean Peninsula is surrounded by four marginal seas: the Yellow Sea, East China Sea, Korea Strait, and East Sea (Sea of Japan). These regions provide distinct habitat conditions for marine life due to their unique ocean dynamics and physical environments, leading to biogeographic division of three ecoregions characterized by different biota (Rebstock & Kang, 2003; Han & Lee, 2020; Spalding et al., 2007). The western coast of Korea is strongly affected by the Yellow Sea. It generally shows low salinity in summer and high salinity in winter (Spalding et al., 2007). The general characteristic of the southern coast of Korea is a relatively warm current with high salinity under the influence of the Tsushima Current of the Korea Strait which branches out of the East China Sea (Lim et al., 2019). The East Sea is characterized by low temperature and high salinity, which is known to have an oceano-geographic intersection where warm and cold currents meet. Naturally, barnacles inhabiting Korea show species-specific geographic distribution based on their preference for such ocean characteristics (Kim et al., 2020). Kim et al. (2020) collected 20 species of native and introduced barnacles in 44 localities of Korea from 2016 to 2018 and tracked changes in their distribution along with water temperature and salinity changes. They found that A. eburneus populations had established in the southwestern coast of Korea representing the East China Sea ecoregion. They also found that A. eburneus preferred relatively warm water over the cooler water in both northeastern and northwestern parts of Korea.

Recently, A. eburneus emergence was noted in an ecological monitoring project in the Korean harbors (“Improvement of management strategies on marine ecosystem disturbing and harmful organisms”). Underway since 2013, the project is aimed at monitoring various alien marine organisms invading the Korean harbors through ship fouling or ocean currents. During a survey in April 2021, extensive A. eburneus spread was documented in four major harbors, namely Incheon, Tongyeong, Sokcho, and Hanlim, representing all surrounding Korean waters, including the Yellow Sea, East China Sea, Korea Strait, and East Sea (Sea of Japan). Considering the substantial geographic distances and heterogeneous habitat conditions among these regions, we speculated that A. eburneus might exhibit adaptive genetic and morphological variation across geographies and possible routes of invasion.

To this end, in the present study, we first addressed the question of genetic variability by analyzing the genetic structure of Korean A. eburneus populations using a mitochondrial genetic marker. Then, we examined variations in the shape and left-right symmetry of the opercular plates of A. eburneus, which are of evolutionary and taxonomic significance (Pitombo et al., 2017), to evaluate morphology relative to habitat. For analysis, we employed two-dimensional landmark-based geometric morphometrics (LBGM)—a powerful statistical tool for quantifying morphological variations (Slice, 2007) to distinguish populations. Finally, we estimated the density of A. eburneus individuals to assess potential differences in recruitment among localities.

Materials and Methods

Sample collection and installation of plate

We performed the monitoring survey and sample collection from four localities of the southern Korean Peninsula, Incheon (37°27′41.4″N, 126°36′49.8″E), Tongyeong (34°49′38.1″N, 128°26′03.5″E), Sokcho (38°13′36.7″N, 128°35′19.6″E), and Hanlim (33°25′11.2″N, 126°15′40.1″E) harbors (Fig. 1A). We accessed the ports and made collections with the permission of the Korea Institute of Marine Science & Technology Promotion (KIMST, Busan, South Korea, project number: 20190518). We installed a set of collecting panels consisting of 10 acrylic attachment plates (3 × 3 dm) (Fig. 1B) per location in April 2020, submerged at a depth of 1–3 m from sea level. Plate numbers were placed in order starting with those closest to the surface of the water. The plates were deployed for a duration of 12 months to include exposure across both warm and cold seasons. The warm season lasted from April to October 2020 (spring and summer), while the cold season lasted from November 2020 to March 2021 (fall and winter). We collected respectively 60, 55, 61, and 82 individuals of A. eburneus from the plates of Incheon, Tongyeong, Sokcho, and Hanlim harbors in April 2021. Since the maximum age of adult A. eburneus (Fig. 1C) remains unknown and their lifespan may vary with food availability and environmental factors, we collected only individuals with basal diameters of 2–2.5 cm from the assemblage, the maximum reported diameter for the species (Kaplan, 1988; Gosner, 1999). We did a sample collection at random due to the appropriate size of individuals being unevenly distributed from plate to plate. We photographed each recruitment plate before barnacle collection using a digital camera (TG-5; Olympus, Shinjuku City, Tokyo, Japan) and used a knife to remove the individuals from the plate and immediately preserved them in 95% ethanol. To examine the seasonal changes in water temperature and salinity, we collected these data using a handheld YSI Pro30 temperature and conductivity meter (YSI, Yellow Springs, OH, USA) at the four sampling sites every 3 months from May 2020 to April 2021 (Table S1).

Figure 1 Localities and methods of survey.

(A) Map of the Korean Peninsula displaying collection sites of Amphibalanus eburneus for this study. Blue circle: Incheon; sky blue circle: Tongyeong; Violet circle: Sokcho; grey circle: Hanlim. (B) Photographs of 10 acrylic attachment plates for monitoring. (C) Photograph of A. eburneus in dorsal view.

DNA amplification and genetic analyses

We randomly selected 30 individuals from each locality sample and isolated genomic DNA with the aid of the LaboPassTM Kit (Cosmo, Seoul, Korea) following the manufacturer’s protocols. We amplified partial sequence of the mitochondrial gene cytochrome C oxidase subunit I (COI) with polymerase chain reaction (PCR) using PCR premix (BIONEER. Co, Daejeon, Korea) in an AllInOneCycler™ PCR system (BIONEER. Co, Daejeon, Korea). We used the universal primer pair, jgLCO1490 (5`‒TIT CIA CIA AYC AYA ARG AYA TTG G‒3`) and jgHCO2198 (5`‒TAI ACY TCI GGR TGI CCR AAR AAY CA‒3`) (Geller et al., 2013) with the following amplification protocol: initial denaturation at 94 °C for 2 min, followed by 30 cycles of denaturation at 94 °C for 1 min, annealing at 48 °C for 1 min, extension at 72 °C for 1 min, final extension at 72 °C for 10 min, and storing at 4 °C. We purified the PCR products for sequencing reactions using the Labopass PCR Purification Kit (Cosmo, Seoul, Korea) following the instructions of the manufacturer. We sequenced DNA on an ABI automatic capillary sequencer (Macrogen, Seoul, Korea) using the same set of primers.

We confirmed sequence identities with BLAST search (Altschul et al., 1990), and visualized using Finch TV, version 1.4.0 (https://digitalworldbiology.com/FinchTV) to check the quality of signal and sites with possible low resolution. We deposited all obtained sequences in GenBank (OM060406–060438; OK103580–103597) and performed a population genetic analysis at the continental scale, comparing the level of pairwise genetic differentiation between all obtained sequences. We performed sequence alignment using the MAFFT v7.313 (Katoh & Standley, 2013) and included 11 additional sequences of A. eburneus already published and publicly available: four from Tangier Sound (North Atlantic Ocean, MK308058, MK308095, MK308188, MK308249), four from Mastic beach (North Atlantic Ocean, MZ595234, MZ595235, MZ595236, MT192780), and three from Gomishan Wetland (Caspian Sea, MK240317, MK240318, MK240319). We estimated uncorrected pairwise distances using Geneious prime (https://www.geneious.com/prime/) and used the TCS algorithm implemented in PopART (Clement, Posada & Crandall, 2000; Leigh & Bryant, 2015) to evaluate genealogical relationships among COI haplotypes by reconstructing a haplotype network. We used DnaSP 5.10 (Librado & Rozas, 2009) to estimate haplotype diversity (Hd) (Rozas & Rozas, 1999), nucleotide diversity (π) (defined as the average number of pairwise nucleotide differences, and their standard deviations) (Tajima, 1983; Nei, 1987). We examined the degree of gene flow among populations using Arlequin 3.5 (Excoffier & Lischer, 2010) based on parameters FST (Hudson, Slatkin & Maddison, 1992) to check phylogeographic structure among populations. It ranges from little genetic differentiation among populations (0–0.05) to great genetic differentiation (>0.25) (Wright, 1943; Graves & McDowell, 2003). We performed an Analysis of Molecular Variance (AMOVA, Netphen, Nordrhein-Westfalen, Germany) to investigate the contribution of variance components (based on pairwise genetic/Euclidean distances) of gene frequencies in different population levels relative to the total variance (Excoffier, Smouse & Quattro, 1992). We defined three groups (Korean, Caspian Sea, North Atlantic Ocean) accroding to geographic regions for seven populations (Incheon, Tongyeong, Sokcho, Hanlim, Gomishan Wetland, Tangier Sound, Mastic beach) of sequences to measure the amount of genetic variance that can be explained by population structure based on F-statistics (Wright, 1965).

Morphological data acquisition

We dissected the animals under a stereo dissecting microscope, Nikon SMZ 1000 (Nikon, Tokyo, Japan). For each individual, we first separated the left and right sides of the opercular plates from the body and subsequently divided them into scutum and tergum. We used 5% sodium hypochlorite (NaClO) to clean the surface of dissected parts. We placed the dissected parts on a petri dish covered by black paper and included a scale to take calibrated microscope images. To make a dorsal perspective angle perpendicular to the microscope objective, we manipulated the specimen using forceps. After each specimen was adequately positioned consistently, we took images on ventral view of scutum and tergum in multiple foci using a camera DP22 (Olympus, Tokyo, Japan) implemented in the dissecting microscope. To illustrate the three-dimensional depth of field more fully, we used stacking software Helicon Focus 7.7.5 (Kozub et al., 2008) to combine images.

We generated two TPS files for the scutum (148 individuals) and the tergum (151 individuals) separately to evaluate the geometric variation in size and shape, including the asymmetry in side-by-side pairs (matching symmetry between left and right parts) (Ho et al., 2009). We generated copies of all images and employed them along with originals for producing TPS files using tpsUtil software (Rohlf, 2015). We digitized the chosen landmark (LM), all Type I (Bookstein, 1991), twice using tpsDig2 software (Rohlf, 2010) to estimate digitization-related errors (Klingenberg, Barluenga & Meyer, 2002). We selected five anatomical reference points of the scutum for LM digitization (Fig. 2A): one located on apex (LM 1); one located on the posterolateral tip (LM 2); one located along the inflection point of basal margin (LM 3); one located along the inflection point of medial margin (LM 4); one located on the lower point of basal ridge (LM 5). We selected ten anatomical references of the tergum for LM digitization (Fig. 2B): one located on apex (LM 1); one located along the inflection point of lateral margin (LM 2); one located on the distolateral tip of scutal side (LM 3); one located on the basal and spur margin intersection on scutal side (LM 4); one located on the spur distolateral point (LM 5); one located on the spur distomedial point (LM 6); one located on the basal and spur margin intersection on carinal side (LM 7); one located along the inflection point of basal margin (LM 8); one located on the distomedial tip of cranial side (LM 9); one located on the proximomedial tip of cranial side (LM 10). The scutum and tergum datasets finally included 296, and 302 digitized images (from the original 148 and 151 images), respectively.

Figure 2 Opercular plates of Amphibalanus eburneus.

(A) Ventral view of the left scutum with anatomical references for landmark digitization (marked by blue circles). (B) Ventral view of the left tergum with anatomical points for landmark digitization. Scale bar: 1 mm.

Geometric morphometric analyses

We employed algorithms implemented in Morpho J package software ver. 1.07d (Klingenberg, 2011) for all LBGM analyses. We aligned and superimposed all landmark configurations in six TPS files with Generalized Procrustes Analysis (GPA) to remove the effects of non-shape variation (Rohlf & Slice, 1990). We converted the Procrustes shape coordinates into a covariance matrix (Brusatte et al., 2011). As a size proxy, we estimated the centroid size (CS) for each individual from the raw LM coordinates (Bookstein, 1989). We calculated the CS as the square root of the sum of squared distances for a set of centroid LMs (Mitteroecker et al., 2013). We performed Procrustes analysis of variance (ANOVA) test for group structuring evidence in the overall dataset using population and side as classifiers. We also used it to evaluate digitizing errors (Klingenberg & McIntyre, 1998). After implementing the Procrustes ANOVA to test error terms, we employed the first digitization dataset and divided it into the left and right datasets. We carried out typical LBGM analyses on the left and right datasets, including regression, principal component analysis (PCA), and canonical variate analysis (CVA), following the procedure previously described in Kim et al. (2021). We performed regressions of shape onto size to test allometry based on regression scores and CS (Monteiro, 1999; Klingenberg, 2016). We applied a permutation test (Good, 2013) to assess the statistical significance against the null hypothesis. We estimated residual components to subtract the portion of shape variations predicted by the regression for further analyses. We used the residual shape component for PCA, which is frequently applied for the first exploratory analysis of a large dataset composed of several samples to provide a visual impression of overall shape variations (Mitteroecker et al., 2013). We used the wire frame to visualize scutum and tergum’s average shape variations along major PCA axes. We employed separated residual components for CVA, which is a multivariate method, producing a criterion for reliably distinguishing among multiple groups preliminarily defined. The analysis generated a multivariate statistical value as Mahalanobis Distances, (MD) (Timm, 2002). The permutation test assessed the statistical significance against the equal group means’ null hypothesis.

Measurement of density of recruited barnacle

We calculated the density of A. eburneus per attachment plate across sampling areas as the number of barnacles found in each plate photograph by its area (9 dm2). We counted the individual number of A. eburneus based on pixel analysis of photographs using ImageJ software (Schneider, Rasband & Eliceiri, 2012). We estimated the mean, standard deviation, maximum, and minimum values of density.

Results

Population genetic diversity

Fifty-seven COI patrial sequences were obtained: ten from Incheon, 12 from Tongyeong, 13 from Sokcho, 22 from Hanlim. The final alignment including GenBank sequences comprised 68 sequences, which included no stop codons, and the sequences encoded polypeptides of 208 amino acids. The alignment was trimmed to a length of 627 base pairs, of which 579 were constant and 48 were variable (16 singleton and 32 parsimony informative). The average nucleotide frequency of the aligned sequences was 28.8%, 37.4%, 16.1%, and 17.7% for each A, T, G, and C, respectively. In other words, the sequences were AT-rich (66.2%). The uncorrected pairwise distances between the four populations ranged between 0 and 3.5% (Table S2), with the highest value recorded between the individuals of Tongyeong and Hanlim.

The parsimony network generated with TCS (Fig. 3) detected 33 haplotypes (Hd = 0.961, π = 0.016) from all obtained sequences, forming three distinct clades (A, B, and C). Clade A was separated from clade B by 12 mutational substitutions. Clade C was distinguished from clades A and B by 12 and six substitutions, respectively. Overall, the frequency of haplotypes in each clade was not locality-specific but clade A contained only individuals matching closely with those from the Atlantic Ocean and Caspian Sea, clade B contained only indivduals matching closely with those from the Caspian Sea, whereas clade C individuals were intermediate with no identifiable match with any sea or ocean. Nearly all haplotypes detected presented widespread geographic distribution except for those of Tangier Sound, which were only found in clade A. The Incheon population included seven haplotypes (Hd = 0.933, π = 0.01336) belonging to clades A and B. Four of these seven haplotypes were shared by Tongyeong, Hanlim, Tangier Sound, and Mastic beach populations. The Tongyeong population comprised eight haplotypes (Hd = 0.924, π = 0.01418) belonging to clades A and B. Two of these eight haplotypes were shared by Sokcho, Hanlim at the former clade. The Sokcho population included eight haplotypes (Hd = 0.897, π = 0.01795), two of which exclusively belonged to clade C. The Hanlim population included 14 haplotypes (Hd = 0.957, π = 0.01668), which belonged to clades A and B.

Figure 3 TCS haplotype network generated using 68 mtCOI sequences of Amphibalanus eburneus collected from four sites in Korea.

Different colors and alphabets in each circle indicate different collecting sites. Sizes of nodes and pie segments are proportional to haplotype frequency. Vertical parallel lines of the network represent the number of substitutions.

Encompassing all haplotypes of A. eburneus, the evidence of population genetic structuring in Korea was not statistically significantly different as revealed by pairwise Fst analysis (Table 1). However, Incheon, Tongyeong, and Sokcho populations of Korea were differentiated from North Atlantic populations, Tangier sound, and Mastic beach, with Fst values ranging from 0.41849 to 0.52013. While the Caspian population, Gomishan wetland, was differentiated from North Atlantic ones, with Fst values of 0.29225 for Tangier Sound and 0.30524 for Mastic Beach. In the total genetic variance among groups and populations based on structure, 27% was attributed to the group difference, and 72.3% was explained by the individual differences within populations (Table 2).

Table 1 Estimation of gene differentiation (Fst) values of total populations.

	Incheon	Tongyeong	Sokcho	Hanlim	Tangier sound	Mastic beach	
Tongyeong	−0.05143						
Sokcho	−0.01223	0.04245					
Hanlim	0.03482	0.07402	0.00013				
Tangier sound	0.48938*	0.52013*	0.41849*	0.19610*			
Mastic beach	0.48234*	0.51564*	0.26840	0.18695*	0.00000		
Gomishan wetland	0.11523	0.15744	0.27339	−0.02264	0.29225*	0.30524*	
Note:

* Indicates statistical significance at p < 0.05.

Table 2 Results of the AMOVA analyses of seven populations of A. eburneus with grouping according to geographic regions, Korea, North Pacific Ocean, and Caspian Sea.

Variation source	d. f.	Sum of square	Variance components	Variation percentage	F-statistic	
Among groups	2	42.245	1.70608	27	Fct = 0.26931*	
Among populations within groups	4	20.419	0.04612	0.7	Fst = 0.27659*	
Within populations	61	279.557	4.58290	72.3	Fsc = 0.00996*	
Total	67	342.221	6.33510			
Note:

* Indicates statistical significance at p < 0.05.

Variations in size and shape

The ANOVA results (Table 3) yielded negligible digitizing errors for all datasets, with the individual variability mean square (MS) and F values far exceeded the error values. The individuals significantly varied in terms of size of the scutum (F = 7.99, p < 0.0001) and tergum (F = 3.58, p < 0.0001). However, neither population nor asymmetry side-by-side contributed to the observed variations in the size of the scutum (asymmetry by the side, p = 0.7304) and tergum (population, p = 0.1104; asymmetry by the side, p = 0.194). The effect of asymmetry side-by-side was significant, contributing to the variations in the shape of the scutum (F = 5.49, p < 0.0001) and tergum (F = 6.33, p < 0.0001) and exceeding that of individual variability. The population most significantly contributed to the variations in the shape of the scutum (F = 7.39, p < 0.0001) and tergum (F = 8.23, p < 0.0001), and its contribution exceeded that of the asymmetry by the side.

Table 3 Variation in the size and shape of the scutum and tergum inferred by Procrustes ANOVA using a randomized permutation procedure (10,000 iterations).

Part	Factor	SS	MS	df	F	p	
Scutum	Size	
Population	27.982413	9.327471	3	2.92	0.0414	
Individual	181.80021	3.134486	58	7.99	<0.0001	
Side	0.091607	0.091607	1	0.12	0.7304	
Digitizing	0.003736	0.000032	117	5.29	0.1721	
Shape	
Population	0.25211324	0.0140062911	18	7.39	<0.0001	
Individual	0.65532580	0.0018831201	348	1.88	<0.0001	
Side	0.03222771	0.0053712850	6	5.49	<0.0001	
Digitizing	0.00123864	0.0000017644	702	0.78	0.7787	
Tergum	Size	
Population	14.863276	4.954425	3	2.09	0.1104	
Individual	146.846511	2.368492	62	3.58	<0.0001	
Side	1.139178	1.139178	1	1.72	0.1940	
Digitizing	0.10168	0.000776	131	7.79	0.2792	
Shape	
Population	0.31311889	0.0065233103	48	8.23	<0.0001	
Individual	0.78594949	0.0007922878	992	3.53	<0.0001	
Side	0.02271775	0.0014198593	16	6.33	<0.0001	
Digitizing	0.00117952	0.0000005627	2096	0.78	0.7956	
Note:

SS, sum of squares; MS, mean squares; df, degrees of freedom; F, Goodall’s F critical value; p, probability of finding a random value larger than the observed value.

Allometry and size-corrected shape variations among populations

Regression analysis revealed an allometric effect in the left (4.33%, p = 0.011) and the right (3.6%, p = 0.0459) scutum datasets, thus the null hypothesis regarding isometric shape development was rejected. PCA based on residuals revealed major shape variations of the left (Fig. 4A) and right scutum (Fig. 4B), with the first two axes explaining respectively 69.6% (PC1 = 37.5%; PC2 = 32.1%) and 66.5% (PC1 = 41.3%; PC2 = 25.2%) of the total variance. Although PCA on both datasets did not reveal apparently distinguishable clustering among the populations, the Incheon population was recognizable in the PC1 morphospace of the left scutum based on a slightly different trend. Specifically, the Incheon individuals occupied the space between −0.15 and 0.05 for the left scutum, with a negative center of gravity. The wireframe demonstrated the shape variations in Incheon individuals corresponding to PC1. As such, the left (Fig. 4C) and right scuta (Fig. 4D) were horizontally narrower than the average due to medially shifted LMs 1–4 and LMs 1–3, respectively.

Figure 4 Principal component analysis of residuals of the scutum shape coordinate.

(A) Scatter plot of the left scutum. Principal components 1 and 2 are indicated on x-axis and y-axis, respectively. (B) Scatter plot of the right scutum. Blue squares: Incheon individuals; sky blue circles: Tongyeong individuals; violet triangles: Sokcho individuals; grey pentagons: Hanlim individuals. (C) Wireframes of shape change in the left scutum (black line) corresponding to PC score against the mean shape (grey line). (D) Wireframes of shape change in the right scutum.

Regression analyses rejected the allometric association in the left (1.983%, p = 0.127) and right (1.597%, p = 0.3037) tergum datasets. The first two PC axes of the left (Fig. 5A) and right tergum (Fig. 5B) datasets explained respectively 38.9% (PC1 = 23.7%; PC2 = 15.2%) and 39.3% (PC1 = 24.6%; PC2 = 14.7%) of the total variance. The PC1 of both datasets emphasized the variations in the shape of the tergum among the populations, with the morphospace clustering of the Tongyeong population on one side (with the center of gravity in the negative part of the PC1). The Sokcho population was clustered on the negative side in the left tergum dataset but not in the right one. Meanwhile, the Incheon and Hanlim populations displayed clustering on the positive side of PC1 morphospace. The wireframe corresponding to PC1 (Figs. 5C and 5D) showed a narrow and elongated shape for the Tongyeong and Sokcho populations due to proximally shifted LMs 1, 2, and 8; medially shifted LMs 3, 4, 9, and 10; and distally shifted LMs 5, 6, and 7.

Figure 5 Principal component analysis of residuals of the tergum shape coordinate.

(A) Scatter plot of the left tergum. Principal components 1 and 2 are indicated on the x-axis and y-axis, respectively. (B) Scatter plot of the right tergum. Blue squares: Incheon individuals; sky blue circles: Tongyeong individuals; violet triangles: Sokcho individuals; grey pentagons: Hanlim individuals. (C) Wireframes of shape change in the left tergum (black line) corresponding to the PC score against the mean shape (grey line). (D) Wireframes of the shape change in the right tergum.

Population differentiation based on morphological distance

Based on the Mahalanobis distances (Table 4), the Tongyeong population was the most distantly related to the rest, with the highest values of the left and right tergum morphology. The permutation test of CVA rejected the PE null hypothesis for equal group means between populations (p < 0.0001). CVA of the left and right tergum datasets showed that the first two axes explained respectively 81.4% (CV1 = 42.7%; CV2 = 38.7%) and 88.5% (CV1 = 69.6%; CV2 = 18.9%) of the total variance.

Table 4 Comparison of variation in the mean shape of the scutum and tergum among populations using canonical variate analysis (CVA).

	Locality	Incheon	Tongyeong	Sokcho	
Left tergum	Tongyeong	3.9417/0.0001			
Sokcho	3.7791/0.0001	4.3321/0.0001		
Halim	2.5423/0.0001	4.3213/0.0001	3.4708/0.0001	
Right tergum	Tongyeong	4.2982/0.0001			
Sokcho	2.7491/0.0001	5.3365/0.0001		
Halim	3.2629/0.0001	5.7166/0.0001	2.6632/0.0001	
Note:

Left score, Mahalanobis distance; Right score, probability of finding a random value larger than the observed value.

Plate density of Amphibalanus eburneus

Table 5 summarizes plate density of A. eburneus recruitment in the surveyed areas. The Incheon population showed the most significant mean density at 65.27 indi/dm2, followed by Hanlim, Sokcho, and Tongyeong populations (density at 54.89, 32.04, 8.67 indi/dm2, respectively).

Table 5 Density of recruited A. eburneus individuals in the attachment plate of all surveyed areas.

Plate no.	Incheon	Tongyeong	Sokcho	Hanlim	
1	28/252	13.11/118	3.11/28	56/504	
2	36.11/325	18.44/166	1.56/14	71.33/642	
3	65.11/586	15/135	1.44/13	59.78/538	
4	67.67/609	9.78/88	8.89/80	82.44/742	
5	73.67/663	8.33/75	3.11/28	38.67/348	
6	85.22/767	5.44/49	14.44/130	46.44/418	
7	61.11/550	1.33/12	33.56/302	49.89/449	
8	88.22/794	6.89/62	71.33/642	42/378	
9	71.22/641	2.78/25	75.78/682	59.44/535	
10	76.33/687	5.56/50	107.22/965	42.89/386	
Mean	65.27	8.67	32.04	54.89	
SD	19.48	5.18	36.74	13.21	
Note:

Left score, density; Right score, individual number; Plate No., plate number; SD, standard deviation; Bold characters, maximum and minimum values estimated per locality.

Discussion

The present study investigated geographical differentiation in the genetics, morphology, and recruitment of Korean populations of alien barnacle A. eburneus. Based on the pairwise distance of the COI sequences, A. eburneus populations showed a low diversification rate. The values ranged between 0% and 3.5%, falling within the level of intra-specific variability for this species, compared to the much higher inter-specific values of COI sequences in other balanomorph barnacles (Tsang et al., 2008; Chen et al., 2014). Parsimony haplotype network analysis revealed detailed genetic characteristics of the populations, establishing three separate clades. Interestingly, the clade separation was not locality-specific, and nearly all haplotypes of the populations were randomly placed in the three clades. Fst value comparison confirmed an absence of genetic structuring between Korean populations indicating A. eburneus has been introduced in all directions of the Korean Peninsula without significant genetic differentiation suggesting that dispersal and/or delivery are intermixed. As such A. eburneus’s invasion success could be primarily attributed to its wide range of adaptability in various salinity and water temperatures (Costlow & Bookhout, 1957; Dineen & Hines, 1994), comparable to other introduced balanomorph species. Amphibalanus improvisus (Darwin, 1854) has been found to exhibit a complex haplotype admixture with global distribution (Chen et al., 2014; Wrange et al., 2016) based on a high degree of euryhalinity and eurythermy (Pansch, Schlegel & Havenhand, 2013). It showed a similar distribution in the Korean Peninsula (Kim et al., 2020). Contrarily, another balanomorph invasive species, Perforatus perforatus (Bruguière, 1789), showed a distribution confined to eastern and southern parts of the Korean Peninsula owing to its weak capability to vary physically (Kim et al., 2020). Indeed, the capability to endure environmental fluctuation is known as a significant biological factor determining various marine invertebrates’ geographical distributions (Chang et al., 2017; Seo et al., 2021).

Parsimony network analysis revealed that Korean populations shared haplotypes with North Atlantic and the Caspian Sea populations. The genetic structure indicated by Fst value showed that there was genetic differentiation only between Korean and North Atlantic populations. This corresponded to the AMOVA results, where within-population variations accounted for the majority of the total variation. Our results might indicate a possible genetic relationship between the Korean and Caspian populations, although the latter region is geographically isolated at a large distance from the Far East. However, it is difficult to confirm such a hypothesis clearly due to extremely small numbers of sequence in the Caspian Sea population, which did not allow for establishing the overall genetic structure of both regions. Consequently, it is also challenging to speculate on the actual invasion pathway into the Korean Peninsula from our results owing to insufficient genetic information. Shipping and other anthropogenic activities might have played an important role in shaping the current population genetic structure of A. eburneus. Indeed, fouling organisms including barnacles often form dense populations on ship hulls as large founding populations and/or potential admixture, which is necessary for the invaders to overcome the founder effect and demographic bottlenecks (Roman & Darling, 2007; Dlugosch & Parker, 2008). Previous monitoring surveys (Park et al., 2017; Kim et al., 2020) support our speculation that Korean A. eburneus populations are more frequently found in ports than in exposed habitats. It is also likely that shipping along the coastline is more responsible for the gene flow among Korean populations than internal shipping. Because internal shipping across the Korean Peninsula is impossible due to the mountain range existing along the eastern half, disturbing direct linkage between the east and west coast.

Our LBGM analysis revealed substantial geometric variations in the opercular plates of A. eburneus, which are reported for the first time in the study of barnacles. We identified two different aspects of morphological variations in the datasets: allometry and shape variations between sides and among populations. Allometry is a known factor contributing to morphological integration (Klingenberg, 2013, 2016), and group discrimination can often be improved following size correction (Sidlauskas, Mol & Vari, 2011). The examined individuals significantly varied in terms of the size of the scutum and tergum, indicating moderate allometric effects on the scutum morphology. Although we could not markedly improve population discrimination following size correction, the discrepancy in the presence of allometry between the scutum and tergum is noteworthy. In recent years, with increase in the number of studies analyzing invertebrate morphology using LBGM, cases of independent evolution of specific body parts, regardless of being physically connected to one another have been noted (Karanovic & Bláha, 2019; Karanovic, Huyen & Brandão, 2019; Budečević et al., 2021). In this context, our findings suggest that the scutum and tergum of A. eburneus have evolved in size independently, despite being connected to each other to form the opercular plate.

Furthermore, similar trends were noted in the size-corrected shape variations between the sides. As such, the left and right sides of the plates were not described by the identical direction of LM shift (see Figs. 4C, 4D, 5C and 5D). Indeed, asymmetry side-by-side of the opercular plates is not unknown in barnacles. Specifically, members of the order Verrucomorpha, which inhabit deep-sea hydrothermal vents among other habitats, exhibit an asymmetric form of the scutum, with one side movable and the other fixed (Newman, 2000). However, apart from taxa presenting readily apparent disparity, geometric asymmetry between the paired sides of the opercular plate has never been documented in acorn barnacles, with the exception of a previous study in which linear measurements of asymmetry were reported (Barnes & Healy, 1971; Chen et al., 2014). In our dataset, individuals from the Sokcho population showed obvious asymmetry in the tergum, forming a distinct cluster in the left and right morphospace of the PCA biplot (see Figs. 5A and 5B). Therefore, the two sides of the opercular plate have likely traced independent evolutionary paths in terms of shape, which may lead to incorrect population or species identification depending on which of the two sides is selected for taxonomic examination.

Regarding population differentiation, the Tongyeong population could be clearly differentiated from the rest based on variations in the shape of the tergum on both sides of the opercular plate. Several external factors that may cause this phenomenon. High recruitment density can influence barnacle shape, often resulting in columnar shells in response to intensive competition for the space and foraging (Barnes & Powell, 1950; Bertness, Gaines & Yeh, 1998; Hills & Thomason, 2003). Our analysis, however, did not suggest a relationship between shape variation and intra-specific competition since the Tongyeong population showed the lowest recruitment densities. According to Barnes & Healy (1971), water temperature is also another major factor affecting variations in the opercular plate morphology of A. eburneus. Based on linear measurement results, they reported that A. eburneus in cold regions differed markedly from in the warm regions by having the basal margin of the carinal side of the tergum deeply hollowed out. Lively (1986) reported shape variations in the barnacle Chthamalus anisopoma Pilsbry, 1916 exposed to the predatory snail Mexacanthina lugubris angelica Oldroyd, 1918; as such, under predation pressure, some juveniles developed a bent morphology. Jarrett (2008) reported that Chthamalus fissus Darwin, 1854, which possesses an oval operculum, manages predation risk by changing the shape of the plate to become narrower, which is advantageous in escaping the predatory gastropod, Mexacanthina lugubris lugubris Sowerby, 1821. In the present study, we were unable to determine possible correlations of the observed shape variations in Tongyeong populations with annual record of water temperature and salinity due to a lack of significant differences in these parameters with other localities (Table S1). In addition, we did not observe any evidence of predation-related characteristics of the A. eburneus communities in the monitoring plates. Compared with direct observations in natural habitats, surveys using monitoring plates offer limited opportunities to witness natural phenomena representing the relationships among organisms due to limited space and resource accessibility. Nonetheless, based on our genetic analyses, the genetic diversity in the Korean A. eburneus populations imply a great plasticity for adaptation, which can drive variations in shape of the tergum in response to certain external stimuli. Therefore, further studies are warranted on A. eburneus in the adjacent natural habitats using additional information on their community structure and trophic relationships.

Conclusion

Using mitochondrial gene sequence and LBGM analyses, the present study successfully unveiled genetic structure, shell morphology, and recruitment pattern in the alien barnacle A. eburneus in Korea. Genetic comparisons among four regions confirmed three clades based on 3.5% pairwise genetic distances and 33 haplotypes, albeit without regional specificity. In contrast, the quantitatively described variations in opercular plate size and shape did identify a unique convergence in narrow elongated tergum in the population from Tongyeong. This population was also differentiated from all others by reduced recruitment density. Despite the inability to predict major factors driving dispersal and morphological variation in A. eburneus in South Korea, we demonstrated an unique phenotypic form of A. eburneus among diverse genotypes at a single location.

Supplemental Information

Supplemental Information 1 Seasonal records of water temperature and salinity in 2020 and 2021.

Click here for additional data file.

Supplemental Information 2 Uncorrected pairwise distances among the four populations examined using the mtCOI sequences.

Click here for additional data file.

Supplemental Information 3 FASTA file of mtCOI sequence alignments.

Click here for additional data file.

Supplemental Information 4 The XML file of Arlequin software containing Fst and AMOVA analyses results.

Click here for additional data file.

Supplemental Information 5 Scutum and Tergum datasets.

Click here for additional data file.

Supplemental Information 6 Plate photographs of studied sites used for density analysis.

Click here for additional data file.

We would like to thank Jaehyun Kim (Hanyang University) for providing valuable comments on designing the genetic analysis.

Additional Information and Declarations

Competing Interests

Author Contributions

Field Study Permissions

DNA Deposition

Data Availability

The authors declare that they have no competing interests.

Jeongho Kim conceived and designed the experiments, performed the experiments, analyzed the data, prepared figures and/or tables, and approved the final draft.

Michael Ubagan performed the experiments, analyzed the data, prepared figures and/or tables, and approved the final draft.

Soyeon Kwon performed the experiments, analyzed the data, prepared figures and/or tables, and approved the final draft.

Il-Hoi Kim conceived and designed the experiments, authored or reviewed drafts of the article, and approved the final draft.

Sook Shin conceived and designed the experiments, authored or reviewed drafts of the article, and approved the final draft.

The following information was supplied relating to field study approvals (i.e., approving body and any reference numbers):

The harbor access and the field experiment were approved by the Korea Institute of Marine Science & Technology Promotion (project number: 20190518).

The following information was supplied regarding the deposition of DNA sequences:

The mitochondrial cytochrome c oxidase partial sequences are available at GenBank: OK103580 to OK103597; OM060406 to OM060438.

The following information was supplied regarding data availability:

The raw data are available in the Supplemental Files.

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
