# Peer review of "Variation in genetics, morphology, and recruitment of the invasive barnacle Amphibalanus eburneus (Gould, 1841) in the southern Korean peninsula"

_PeerJ, doi:10.7717/peerj.14002_

## Round 0.1 · original submission · Major Revisions

I have heard back from three expert reviewers, who each have provided a long and extensive list of issues for you to address. Of chief concern to me is the low number of specimens analysed via genetic analyses, an issue that two of the three reviewers also mention. Please respond to all comments in a robust manner with clear explanations.

·

Basic reporting

See 4 below

Experimental design

See 4 below

Validity of the findings

See 4 below

Additional comments

This MS examine the molecular and morphological variation in A. eburneus in Korean waters. I would recommend accept after the revision below can be addressed:

1) Line 193 – symmetry of barnacles – please also cite Ho et al. 2009 for the comparison of left and right sides of the opercular plates

Ho, G. W. C., K. M. Y. Leung, D. Lajus, J. S. S. Ng and B. K. K. Chan (2009). Fluctuating asymmetry of Amphibalanus (Balanus) amphitrite (Cirripedia: Thoracica) in association with shore height and metal pollution. Hydrobiologia, 621: 21-32

2) The introduction should include information on different oceanographic systems in Korea and how do marine species distribute among different system in Korea. In line 95, before introducing A. ebruneus, I would suggest introducing the marine ecoregions in Korea, including the Yellow Sea, the South Sea and Jeju, the East Sea. There are published papers that different marine organisms differ in their distribution among these system, this includes Thoracican barnacles (Kim et al 2020), Rhizocephalan barnacles (Jung et al. 2019, 2021), mussels (Lee et al. 2021). Then go to introduce the invasive barnacle Amphibalanus eburneus. Please cite the following reference and add in my suggested information in the introduction.

Kim HK, Chan BKK, Lee S-k, Kim W (2020). Biogeography of intertidal and subtidal native and invasive barnacles in Korea in relation to oceanographic current ecoregions and global climatic changes. Journal of the Marine Biological Association of the United Kingdom 100, 1079–1091. https://doi.org/10.1017/S0025315420001009

Lee Y., G. Ni, J. Shin, T. Kim, E. M. A. Kern, Y. Kim, S. C. Kim, B. K. K. Chan, R. Goto, T. Nakano, J.-K. Park (2021). Phylogeography of Mytilisepta virgata (Mytilidae: Bivalvia) in the northwestern Pacific: Cryptic mitochondrial lineages and mito-nuclear discordance. Molecular Phylogenetics and Evolution, 157, 107037.

Jung J, Yoshida R, Kim W. 2019. Diversity of parasitic peltogastrid barnacles (Crustacea: Cirripedia: Rhizocephala) on hermit crabs in Korea. Zool Stud 58:33. doi:10.6620/ZS.2019.58-33.

Jung J, Yoshida R, Lee D, Park J. 2021. Morphological and molecular analyses of parasitic barnacles (Crustacea: Cirripedia: Rhizocephala) in Korea: preliminary data for the taxonomy and host ranges of Korean species. PeerJ 9:e12281 https://doi.org/10.7717/peerj.12281

3) For the molecular analysis, I would recommend to add in all available A. eburneus from the Genbank or Barcode of life database, so you can compare the haplotype in Korea with the molecular information outside Korea.

4) The author found that Tongyeong population had different tergum shape than other sites. How about the density and substrate of A. eburneus from Tongyeong? Are they very densely populated or they are solely from very vertical seawall rather than other habitats on other sites? This can be add in the discussion. Need to comment on the density and habitats (vertical walls or inclined walls or rocks) in discussion of opercular plate morphology.

5) In general, many balanid species have high tolerance to environmental conditions and result in very wide distribution and with mixed haplotypes. The lack of genetic differentiation of A. eburneus is probably its high tolerance to salinity and low temperature stress, so it can be present in the Yellow Sea and East Sea. There are also other fauna that have high haplotype diversity, but yet the distribution of these haplotype in Korea are not distinct geographically. Another similar example can be documented in Fistulobalanus albicostatus which there are mixed haplotypes in a northern clade that cover Korea and Japanese waters (Chang et al. 2017). Another non-barnacle example is the jelly fish that have mixed haplotype but distributed over different ecoregions in Korea (Seo et al 2021). I think the authors should also cite this reference as a supporting example that comparable to your study. Other invasive species like A. improvisus also has similar distribution with A. eburneus (see Figure 7 in Kim et al. 2020 see above). Please cite the suggested example and references.

Chang, Y. W., J. S. M. Chan, R. Hayashi, T. Shuto, L. M. Tsang, K. H. Chu and B. K. K. Chan* (2017). Genetic differentiation of the soft shore barnacle Fistulobalanus albicostatus (Cirripedia: Thoracica: Balanomorpha) in the West Pacific. Marine Ecology-An Evolutionary Perspective, 38(2): e12422.

Seo Y, Muhammad BL, Chae J, Ki JS. 2021. Population genetic structures of the jellyfish Aurelia coerulea polyps along Korean coasts and implications as revealed by Mitochondrial COI. Zool Stud 60:63. doi:10.6620/ZS.2021.60-63.


In contrast to A. eburneus, the distribution of the other invasive barnacle Perforatus perforatus which are only limited to East Sea and southern coast of Korea (see updated distribution in Kim et al. 2020, see above), probably due to its lack of tolerance on low salinity waters in the Yellow Sea. The distribution of the invasive barnacles should be compared in the discussion.

·

Basic reporting

The quality of English is very high but a small number of technical corrections is needed.
Literature references need to be enhanced with information on shipping and/or ocean current patterns.

Experimental design

Greater genetic sampling is needed.

Validity of the findings

Further discussion of invasion pathways, an objective of the study, needed.

Additional comments

Dear authors
I found your study of A. eburneus in South Korea novel and interesting but do not find it ready for publication yet. My understanding is that your objectives were to 1) determine if the barnacle was morphologically different among 4 different regions of S. Korea as a consequence of local adaptation, 2) determine if the barnacle was genetically variable among the same 4 regions, and 3) speculate on invasion pathways within Korea. Unfortunately, I think your sample size is too small to adequately address objectives 2 and 3. Your findings of morphological variability are wonderful and I think they could stand on their own as a paper, but with 78% haplotype variability among 18 individuals the possibility of detecting genetic patterns, which you did not, is small. Deeper sampling might reveal a few important rare haplotypes with explanatory power. The number is also too few to assess invasion pathways. I am recommending rejection of the paper in its current form but strongly encourage you to substantially revise and resubmit. Your findings on barnacle shell asymmetry certainly deserve to be published. Below I have summarized my main criticisms followed by a few detailed comments. I enjoyed reading your fine use of language but I have also provided edits on a PDF version of the manuscript to help with any revisions.

My primary criticism is that sample size is too small. I was left wondering why when you collected 248 specimens were only 18 sequenced? Was their high failure in PCR? If so that is a possible indication of variability in the primer region. Regardless, further explanation is needed for justification of the sample size.

Sample size also limits your ability to analyze invasion pathways. Your discussion of the pathways seemed too vague. You seemed to suggest natural dispersal is responsible for the spread in Korea but couldn’t it also be internal shipping? I was looking for some information on shipping patterns or currents or both. Maybe pointing out which harbors are most active and what the predominant shipping pathways and currents are. This needs much further discussion. Much greater sampling will probably make this explanation easier.

Your finding of haplotype group A and B is interesting but what does it mean? Does it point to separate invasions? Is it real? More samples will help.


Abstract
Line 38: Instability of what sort? Environmental? Population?

Detailed questions
Introduction
Line 110: B. eburneus does not have direct development (meaning no larval stage) so I'm not sure what meaning is intended.

122-123: At the end of this sentence do you mean what in their evolutionary history contributes to their invasive ability or do you mean to ask if their invasive ability has changed over time? I think it might be neither but instead you want to identify the route of arrival in Korea. Needs some clarification.

Results
247: Why so few sequences? 18 out of a possible 248 samples right? PCR failure?

Discussion
336-339: this sentence is confusing. Long-term gene flow sounds continuous so multiple times doesn't make sense. Do you mean there were mutliple invasion events? Please clarify.

339: Is it an assumption that A. eburneus invaded in the south first or do you know that? Ships could have brought the invaders to any starting point arbitrarily right?

345-348: And role of ship traffic too. Distributions may not be entirely from natural dispersal right?

355: I'm not sure what is mean't by integration.


Potential references to include:
Allen, B.M., Power, A.M., O'riordan, R.M., Myers, A.A. & Mcgrath, D. (2006) Increases in the abundance of the invasive barnacle Elminius modestus Darwin in Ireland. Biology and Environment: Proceedings of the Royal Irish Academy, 106, 155-161
Ashton, G., Davidson, I., Geller, J.B. & Ruiz, G.M. (2016) Disentangling the biogeography of ship biofouling: Barnacles in the Northeast Pacific. Global Ecology and Biogeography, 25, 739-750
Ávila, E., Araujo-Leyva, O.R., Rodríguez-Santiago, M.A. & López-Rosas, H. (2018) Alien barnacle Amphibalanus amphitrite epizoic on two native oyster species in the southern Gulf of Mexico: spatio-temporal variability and current status of its epibiosis. Marine Biology Research, 14, 581-589
Carlton, J.T., Chapman, J.W., Geller, J.B., Miller, J.A., Carlton, D.A., Mcculler, M.I., Treneman, N.C., Steves, B.P. & Ruiz, G.M. (2017) Tsunami-driven rafting: Transoceanic species dispersal and implications for marine biogeography. Science, 357, 1402–1406
Cohen, O.R., Walters, L.J. & Hoffman, E.A. (2014) Clash of the titans: a multi-species invasion with high gene flow in the globally invasive titan acorn barnacle. Biological Invasions, 16, 1743–1756
Connell, J.H. (1955) Elminius modestus Darwin, a northward extension of range. Nature, 175, 954
Frey, M.A., Simard, N., Robichaud, D.D., Martin, J.L. & Therriault, T.W. (2014) Fouling around: vessel sea-chests as a vector for the introduction and spread of aquatic invasive species. Management of Biological Invasions, 5, 21-30
Horikoshi, A. & Okamoto, K. (2005) The first finding of the introduced barnacle Amphibalanus variegatus (Darwin) in Tokyo Bay. Sessile Organisms, 22, 47-50
Kado, R. (2003) Invasion of Japanese shores by the NE Pacific barnacle Balanus glandula and its ecological and biogeographical impact. Marine Ecology Progress Series, 249, 199-206
Kim, H.K., Chan, B.K.K., Lee, S.-K. & Kim, W. (2020) Biogeography of intertidal and subtidal native and invasive barnacles in Korea in relation to oceanographic current ecoregions and global climatic changes. Journal of the Marine Biological Association of the United Kingdom, 100, 1079 - 1091
Matsui, T., Shane, G. & Newman, W.A. (1964) On Balanus eburneus Gould (Cirripedia, Thoracica) in Hawaii. Crustaceana, 7, 141-145
Schwindt, E. (2007) The invasion of the acorn barnacle Balanus glandula in the southwestern Atlantic 40 years later. Journal of the Marine Biological Association of the United Kingdom, 87, 1219-1225
Sylvester, F., Kalaci, O., Leung, B., Lacoursier-Roussel, A., Clarke-Murray, C., Choi, F.M., Bravo, M.A., Therriault, T.W. & Macisaac, H.J. (2011) Hull fouling as an invasion vector: Can simple models explain a complex problem? Journal of Applied Ecology, 48, 415-423
Yamaguchi, T., Prabowo, R.E., Ohshiro, Y., Shimono, T., Jones, D.S., Kawai, H., Otani, M., Oshino, A., Inagawa, S., Akaya, T. & Tamura, I. (2009) The introduction to Japan of the Titan barnacle, Megabalanus coccopoma (Darwin, 1854) (Cirripedia: Balanomorpha) and the role of shipping in its translocation. Biofouling, 25, 325-333
Yorisue, T., Yoshioka, Y., Sakuma, K. & Iguchi, A. (2018) Evaluating the occurrence of cryptic invasions of a rocky shore barnacle, Semibalanus cariosus, between the north-eastern Pacific and Japan. Biofouling

Reviewer 3 ·

Basic reporting

This manuscript analysed COI sequence and morphological variation of an invasive barnacle Amphibalanus eburneus in Korea. This study seems preliminary because number of individuals used for genetic analysis is too small too detect differentiation among populations. In addition, individual sample replication for morphological analyses are pseudoreplication as the individuals were collected from the plate that attached barnacles and other sessile organisms with high density. Authors shoulf address these two points before acceptance.

Experimental design

L153
This manuscript shows only haplotype network for the genetic data. I recommend to conduct basic population genetic analyses, such as Fst, IBD, AMOVA.

L154
Why only ten individuals were selected for the genetic analyses. It seems too small number to detect the genetic differences among populations.

L261-264
Haplotype diversities of Tongyeong and Sokcho are 1. Are they correct? Ten indivisduals are analysed and detected four haplotypes, so Haplotype diversity should not be 1.

L332-333, L409
Population genetic differentiation should be statistically tested.

L396-399
This is very important. Authors collected sample individuals from a plate with high density. Then the morphological variations detected may be derived from the density of organisms on plate. So replication of individuals for morphological analyses seems pseudoreplication.

Validity of the findings

L50-53
Authors sequenced only mtCOI and it is difficult to discuss whether the observed phenotypic variation is genetic basis or not. Please clarify the difference between “local adaptation” and “selection driven by speciation”. It is impossible to reveal the mechanisms (selection or plasticity) of the morphological differences among sites from this study.

L326
How authors judged the genetic diversity is low. Based on the haplotype network, genetic diversity seems high.

Additional comments

L147-149
Is it possible to calculate densities of barnacles and other sessile species on the plate? Such parameters can affect the barnacle morphologies and worth analysing.

Authors surveyed temp and salinity at each site. Why not use these data for the discussion?

Figure 4 and 5
Please add % of explained for each axis.

---

## Round 0.2 · Major Revisions

I have heard back from the same three reviewers again. Two of whom offer some minor comments and corrections, while the remaining reviewer has offered another round of substantial comments. Please take a look at these comments, and revise your work accordingly. While this decision is again a "major revisions", it is clearly not as major as the first round. Indeed, all three reviewers noted the work is much better than the initial version.

·

Basic reporting

The comments were well addressed and the MS can be accepted for publication. Just very minor corrections below:

Experimental design

ok

Validity of the findings

ok

Additional comments

The author Chan BKK was mis-spelled as Chan BK in the references in line 514, 523, 594, 711 and 719. Please correct the author name to Chan BKK in the reference list.

·

Basic reporting

Some issues with English. Detailed edits provided in attached Word document. See detailed comments in Additional Comments below.

Experimental design

Objectives need focusing or reframing. See detailed comments in Additional Comments below.

Validity of the findings

Novel and meaningful.

Additional comments

Dear authors,
I found your manuscript describing the investigation of adaptive responses in the alien barnacle Amphibalanus eburneus in South Korea interesting and informative. I think your concept of looking for changes in the biology of this barnacle across different environmental regions is excellent. However, I feel that for this particular study you may be attempting to include too much. You identify in your title two areas of focus (genetics and morphology), but I find three in your stated objectives (genetics, morphology, and recruitment), and four in your results and discussion (genetics, morphology, recruitment, and community substratum occupancy). Of course, the four link together to present a comprehensive understanding of the biology of this invader but I think you need to decide whether to reduce the focus of the study to just genetics and morphology and save the ecological aspects for a separate study or to try and reframe it with four objectives. With four objectives you will need to include more background on the ecological community structure of the regions. Though I found the level of written English very high, I am recommending the manuscript for major revision to focus or reframe the objectives.

I have listed below some points or questions that need clarification. In addition I have included a Word document with detailed text edits (particularly the abstract and introduction) made using Track Changes to help with revision. Best regards.

Comments and questions

Title: Needs attention. Korea isn’t the only place in which this barnacle is invasive. Three aspects were included in the study, not just genetics and morphology. See possible alternate title in the attached Word document using Track Changes. Since the region of study encompasses the southern portion of the Korean peninsula, the country of South Korea in particular, I wonder if “South Korea” or “southern Korean peninsula” should be used instead of “Korea”.

Abstract: See detailed edits in the attached Word document using Track Changes.

Introduction: See edits in the attached Word document using Track Changes.

Line 54: The use of “invasive” or “introduced” species needs to be carefully considered. An introduced species is one that appears in a new area, and an invasive species is a species that appears in a new area AND causes environmental or economic damage. Most introduced species do not become invasive. Once a species has appeared and is reproducing and perpetuating itself in the area it is considered “established”. The sequence proceeds first with introduction, second with establishment, and third invasiveness (if it ever reaches that point).

Line 78: Do you have a reference you can cite for “industrial and ecological damage”?

Line 81: The wide salinity tolerance citation of Dineen & Hines refers to a different species of barnacle. I recommend replacing it with the following two citations-

Bacon PR. 1971. The maintenance of a resident population of Balanus eburneus (Gould) in relation to salinity fluctuations in a Trinidad mangrove swamp. Journal of Experimental Marine Biology and Ecology 6:187-198.
Jaberimanesh Z, Oladi M, Nasrolahi A & Ahmadzadeh F. 2019. Presence of Amphibalanus eburneus (Crustacea, Cirripedia) in Gomishan Wetland: Molecular and morphological evidence of a new introduction to the southern Caspian Sea. Regional Studies in Marine Science 25:100469.

Line 83: The wide temperature tolerance of this species could be supported by the following reference-

Scheltema RS & Williams IP. 1982. Significance of temperature to larval survival and length of development in Balanus eburneus (Crustacea: Cirripedia). Marine Ecology Progress Series 9:43-49.

Line 103: The use of “settlement” and “recruitment” need careful consideration. Settlement usually refers to the short-term process of individuals attaching to a surface while recruitment usually refers to longer-term survival. In your study, 1 year on the plates should be considered recruitment.

Methods: This section needs a little reorganizing. I think you should begin with giving information about plate deployment before mentioning sample numbers. Some clarification is needed on the number of plates and perhaps the average number of barnacles collected per plate. 258 barnacles from 40 plates averages 6.45. Was just one string of plates placed in each of the four harbors (i.e. a total of 40 plates)?

I can see from figure 1B that 10 plates are together in a string. The manuscript states the plates size is 3x3 dm (30x30 cm) but the plates do not look to be square. Are the dimensions correct? Same question for line number 259 as well.

How many barnacles were sampled from each plate?. Did you collect ALL individuals 2cm or larger in diameter or only a certain number.

Line 153: You don’t need to mention returning the plates to each location unless you continued collecting data that was used for this experiment.

Lines 154-156: When were water temperature and salinity measurements taken at each location? How often over the year?

Genetics:
Line 160: Manuscript states 10 individuals were sampled per locality for DNA testing (= 40) but you indicate that 57 specimens were sequenced (plus 13 sequences from genebank). Please clarify.

Morphology:
Line 198: For the morphology studies you should indicate how many samples were measured and if they were the same individuals used for sequencing.

Results:
Line 273-277: the list of GenBank samples should be moved to the methods section.

Fig. 3 legend says “18 samples” but it should be 57 plus 13 genbank.

Line 289: It seems Clade A matches best with Atlantic plus Caspian Sea, Clade B with Pacific plus Caspian Sea, and Clade C is intermediate with no match identifed.

Table 4: The heading indicates the table reports the “density” of individuals in which case you should indicate in parentheses “(ind./cm2)” or “(ind./dm2)”.
I am unclear about the listing of N, SD, Max, Min, and %.
For “N” I think you mean “plate number” or “plate ID” not “number of plates”.
Mean and “SD” is clear.
“Max” does not seem to be identified. You could bold and indicate “Max in bolded print”
“Min” does not seem to be identified. You could shade and indicate “Min in shaded print”
“%” is not identified. Does this refer to percent occupancy?
Please clarify.

Table 5. If you decide to remove community data the occupancy values for A. eburneus could perhaps be added to Table 4 if recruitment data is used.

Discussion
Line 410: Unclear what “weak capability to physical variation” means.

Line 430-431: Unclear what is meant here about founding populations and admixture. Do you mean barnacles need to mix genetically before they can successfully invade? Please clarify.

Lines 437-438: Do you need this sentence? Is there any reason to suspect barnacles could cross the peninsula? Are there any water channels that cross the Korean peninsula?

Make the point that in Tongyeong there is a phenotypic convergence for narrow elongated tergum despite diverse genotypic origins.

Conclusion: See edits in the attached Word document strengthening conclusions.

Reviewer 3 ·

Basic reporting

L23
“established”, not “settled”

L151
Please add accession numbers here.

L236
Some Genbank sequences were used for the analyses, which should be explained in Materials and methods.

L381
I found no logical explanation to hypothesize the introduction route to Korean waters. Additionaly, only three specimens were sequenced from Caspian Sea, which is too small to estimate the haplotype composition in the population.

Experimental design

L134; L 236
Ten specimens were sequenced from each Korean site but you got 12, 13, 22 sequences from Tongyeong, Sokcho, Hanlim. Please clarify how many specimens you used for the sequencing.

L 237
Genbank sequences from Ashton et al. (2016) were used for the analyses. These sequences (KU204257; KU204298) were obtained from barnacles attached on hull of international tankers so these sequences should be excluded from your analyses.

I found some other sequences that can be used for your analyses (MT192780; MH546064; KH546065; KU695280). If you include these sequences in your analyses, you will have four sequences from Tangier Sound and seven from New York, which means 11 from North Atlantic coast of the US in total. Then by performing AMOVA, it is possible to test whether the North Atlantic populations are the main source for the Korean populations.

L264; Table1
Authors claim that Korean populations were significantly differentiated from other locations but the P-values were not shown. Please show the P-values of the Fst in table 1.

Validity of the findings

L24; L349
This study does not investigate the process of adaptive evolution through invasion or adaptive phenotypic responses.

---

## Round 0.3 · Minor Revisions

While the paper is close to being acceptable, I must agree with the reviewer regarding the Caspian population. If you use a phylogenetic analyses such as a tree and discuss clades, then you can discuss differences (as in lines 460-462), but my understanding of your text (e.g. see lines 595-596, 826-829) is that you are still doing haplotype and statistical analyses of the Caspian population; this is what is inferred from your text. If you are NOT doing this, please rewrite your text to more explicit and careful about what you are comparing. Can you really say statements like lines 830-834 with N=3? You should at least add words like "potentially" or "possibly" to soften your conclusions from small numbers.

Thus, please carefully examine your paper and what you can and cannot say, and frame your conclusions and statements carefully.

Finally, the English is generally well done but there are some small spelling mistakes here and there. Please thoroughly check your paper again.

·

Basic reporting

Overall the manuscript is excellent. It is clear and unambiguous and the English of professional high quality. A few suggestions for improvement are provided in comments on the attached PDF.

Experimental design

I find no faults with the experimental design.

Validity of the findings

The findings are meaningful, original, and interesting.

Additional comments

This is a well executed, valuable, and interesting study. It is well written, and apart from a few minor comments and corrections noted on the attached PDF, is in my view ready for publication. Somewhere, it would be of great value to add the term "local adaptation". Ideally as a key word if possible.

Reviewer 3 ·

Basic reporting

Authors need to consider limitations of the genetic analyses due to the small number of sequences available from Atlantic and Caspian Sea.

Experimental design

no comment

Validity of the findings

no comment

Additional comments

L446-452
It is impossible to detect the genetic differentiation between Caspian Sea and other regions because only 3 sequences are available from Caspian Sea. Similary, only 8 sequences are available from North Atlantic, it means there is only small power to detect the genetic structure from other regions.

I recommend to remove Caspian Sea sequences from Fst and AMOVA analysis. Still it needs cautious on the interpretation of the results because only 8 sequences are available from North Atlantic.

---

## Round 0.4 · accepted · Accept

Thank you for your perseverance; I am happy to move this into production and look forward to seeing the published version. Congratulations!